# Trimethylamine N-Oxide in Relation to Cardiometabolic Health—Cause or Effect?

**DOI:** 10.3390/nu12051330

**Published:** 2020-05-07

**Authors:** Christopher Papandreou, Margret Moré, Aouatef Bellamine

**Affiliations:** 1Institut d’Investigació Sanitària Pere Virgili (IISPV), 43204 Reus, Spain; papchris10@gmail.com; 2Analyze & Realize GmbH, 13467 Berlin, Germany; mmore@a-r.com; 3Lonza LLC, Morristown, NJ 07960, USA

**Keywords:** trimethylamine N-oxide (TMAO), cardiovascular disease, type 2 diabetes, atherosclerosis, cardiometabolic health, cause‒effect relationship

## Abstract

Trimethylamine-N-oxide (TMAO) is generated in a microbial-mammalian co-metabolic pathway mainly from the digestion of meat-containing food and dietary quaternary amines such as phosphatidylcholine, choline, betaine, or L-carnitine. Fish intake provides a direct significant source of TMAO. Human observational studies previously reported a positive relationship between plasma TMAO concentrations and cardiometabolic diseases. Discrepancies and inconsistencies of recent investigations and previous studies questioned the role of TMAO in these diseases. Several animal studies reported neutral or even beneficial effects of TMAO or its precursors in cardiovascular disease model systems, supporting the clinically proven beneficial effects of its precursor, L-carnitine, or a sea-food rich diet (naturally containing TMAO) on cardiometabolic health. In this review, we summarize recent preclinical and epidemiological evidence on the effects of TMAO, in order to shed some light on the role of TMAO in cardiometabolic diseases, particularly as related to the microbiome.

## 1. Introduction

Cardiovascular diseases (CVD) are the leading cause of mortality and disability, especially in old age [1]. In 2015, the prevalence of type 2 diabetes (T2D) was 8.8% with increasing prevalence rates since 1980 [2] (4.7% in 1980 [3]). Coronary artery disease (CAD), myocardial infarction, stroke, heart failure (HF), atrial fibrillation (AF), and chronic kidney disease (CKD) are known to be closely associated with T2D [4,5], as well atherosclerosis [6], all of which are linked to inflammation [6,7].

Atherosclerosis and T2D can also lead to the development of kidney disease [8,9]. The prevalence of CKD is 5–15% in the adult population (13.6% in the US population [9]) and increases with age [10,11], making it one of the major contributors to the disease network around CVD and metabolic syndrome (Figure 1).

In recent years, there is a mounting body of evidence suggesting that the gut microbiota affects a wide range of systemic conditions, including CVD, T2D, and obesity [12,13], and these conditions also have reverse effects on the gut microbiota (Figure 1).

For instance, there are strong associations of atherosclerosis [14,15] and CKD [16] with dysbiosis. It has been reported that a disturbed proinflammatory microbiome will cause an impaired metabolism of cholesterol and lipids [17] and will lead to a reduction and changes in short-chain fatty acids (SCFAs) [18,19], generated by the gut microbiome. This is further supported by the fact that individuals with metabolic syndrome have an increased Firmicutes/Bacteriodetes ratio, as well as a reduced capacity to metabolize carbohydrates to SCFAs [20]. SCFAs in turn are known to protect from cardiovascular damage by maintaining a proper anti/pro-inflammatory balance [19], possibly by a regulatory T cell-dependent mechanism [21]. In addition, intestinal inflammatory responses (via FABP2-induced intestinal permeability) can aggravate atherosclerotic progress [22]. Dysbiosis also deteriorates CKD by making the lower intestines more permeable to microbiota-generated uremic toxins, which in turn will damage renal tissues [23]. In return, high urea concentrations resulting from CKD modify the gut microbiota to promoteuremic toxin-producing bacteria [23]. Therefore, some microbiota-derived products shift the homeostatic metabolic balance towards a low-grade chronic inflammation, a hallmark of metabolic syndrome [24].

Lately, the gut microbiota-dependent metabolite, trimethylamine N-oxide (TMAO), has received increasing attention for its association with CVD and T2D highlighting the importance of gut microbiome-metabolism in cardiometabolic health.

In diseased populations, elevated plasma TMAO levels have been associated with major adverse CVD events [25] and T2D [26,27]. Furthermore, elevated serum TMAO levels have been proposed as a biomarker of metabolic syndrome [28]. However, in these associations, TMAO has been measured among patients with already established diseases, and thus, it is not possible to assess the causality of increased TMAO levels in the etiopathology of these diseases. More recent prospective observational studies following otherwise healthy subjects but at high cardiovascular risk point to a role of high TMAO plasma levels in preventing the development of T2D, observations supported by animal mechanistic studies.

In addition to the rising number of original scientific literature on TMAO, TMAO has also been subject to extensive reviews, regarding its possible role in conditions such as CVD, T2D, or other aspects of cardiometabolic risk. However, current reviews critically evaluating the cause to effect relationship and the nature of TMAO’s association to cardiometabolic diseases are missing [8]. In the present review, we summarize the evidence from observational studies evaluating the association between TMAO and cardiometabolic diseases. We also discuss the possible mechanisms involved based on animal data and recent longitudinal trials in healthy subjects at risk of developing cardiometabolic diseases.

## 2. Where Does TMAO Come From?

TMAO is a small odorless molecule (molecular mass of 75.11 g/mol) formed by the oxidation of trimethylamine (TMA) by the hepatic Flavin-containing monooxygenases (e.g., FMO1-FMO5 in human [29]). TMA is generated in the colon by the gut microbiome from dietary quaternary amines (e.g., phosphatidylcholine/choline [30], L-carnitine [31], betaine, choline [32], dimethylglycine, and ergothioneine), if these are not fully absorbed during digestion within the small intestines. Generally, TMA is absorbed into the blood; alternatively, TMA can also be degraded to dimethylamine (DMA), methylamine, and ammonia within the colon (Figure 2).

However, the latter reactions take place to a lower extent: in rats, choline supplementation only resulted in a negligible increase in DMA concentration, with a much higher rise of TMA. Thus, non-absorbed choline was mostly converted to TMA, absorbed into the circulation, and oxidized to TMAO in the liver [32]. In a rare genetic disorder, the TMA → TMAO conversion is impaired, resulting in the accumulation of the volatile TMA and trimethylaminuria (fish odor syndrome) [33,34]. Metabolic retroconversion of TMAO to TMA by the gut microbiota has been demonstrated by Hoyles and colleagues [35]. In humans, TMAO and TMA are usually excreted in urine. Serum TMAO and TMA levels increased in patients with reduced renal function (haemodialysis patients) [36], showing the importance of renal clearance for TMAO/TMA level regulation.

In addition, TMAO itself has been shown to be directly derived from fish and sea food consumption and can be directly absorbed in the gut [40,41]. Therefore, plasma TMAO levels are influenced by TMA formation and its degradation as well as the secretion rate of TMA, DMA, and TMAO. Perhaps it would be more relevant to consider the TMA/TMAO ratio as a marker rather than just TMAO levels.

Other TMAO producers are described below.

L-carnitine is present mainly in red meat and dairy products [42]. The estimated daily intake in a western type of diet is 100–300 mg [43]; however, vegetarians have considerably lower L-carnitine intakes. L-carnitine can also be synthesized in the body from methionine and lysine but only 25% in the body of omnivores comes from endogenous synthesis [43]. Therefore, dietary supplementation with L-carnitine can contribute to dietary intake, particularly among vegetarians and people with chronic diseases associated with aging [44]. In addition, L-carnitine is recommended to be added to infant formulas because infants lack the ability to synthesize their own L-carnitine. An infant fed formula supplemented with L-carnitine normalized lipid levels prevented the development of metabolic and energy disorders [45].

The relative absorption of dietary L-carnitine is 54–87% by the small intestine, depending on the amount provided in a meal, but lower (14–18%) when more L-carnitine is ingested as a supplement (usually in larger amounts), pointing to a saturable transporter system [46]. The non-absorbed L-carnitine passes on to be fermented by the microbiome (Figure 2).

Independent from its role in producing TMAO [47,48], L-carnitine plays a crucial part in energy generation by its key role as a shuttle of medium and long chain fatty acids into mitochondria, where fatty acids are metabolized by β-oxidation to produce energy in all tissues, including the heart [49]. The healthy heart uses 50–70% of its energy from fatty acids [50]. Therefore, L-carnitine is essential for normal heart functioning [50,51]. In addition, L-carnitine supplementation reduces oxidative stress, inflammation, and necrosis in many tissues including the cardiac tissue [52]. Other roles of L-carnitine include calcium influx regulation, endothelial function, intracellular enzyme release, and membrane phospholipid composition [52]. L-carnitine supplementation was shown to normalize plasma lipid levels in patients with coronary heart disease [53]. In models of brain injury, L-carnitine improved energy metabolism, decreased oxidative stress, and prevented subsequent cell death [54], or provided tissue protection in ischemic stroke models [55]. In patients with acute myocardial infarction, L-carnitine improved survival by reducing infarct size, ventricular arrhythmias, left ventricular dilation, and heart failure (HF) incidence [56]. L-carnitine has also been associated with better insulin sensitivity in diabetics, insulin-mediated glucose uptake and oxidation in normoglycemic subjects [57], and lower risk of T2D in the PREDIMED (Prevención con Dieta Mediterránea) study [58]. The safety of L-carnitine has been demonstrated in many studies. There is no regulatory limitation on the dosage of L-carnitine by the European Food Safety Authority (EFSA), which considers L-carnitine well characterized [45]. The Spanish Agency for Food Safety and Nutrition limits L-carnitine or L-carnitine hydrochloride supplementation to 2000 mg per day and L-carnitine tartrate supplementation to 3000 mg per day [59].

Choline is found in beef/chicken liver, eggs, wheat germ, bacon, and soybeans among other foods [60]. The Food and Nutrition Board of the Institute of Medicine in the USA has established the daily adequate intake to be 550 mg for men and 425 mg for women [61]. The European Food Safety Authority defined a choline daily adequate intake of 400 mg for adults [62].

Choline is an essential molecule, which functions as a precursor for the synthesis of phospholipids, including phosphatidylcholine, a membrane building block and a precursor of the neurotransmitter acetylcholine. Thus, choline is particularly important for fetal brain development and maintenance of neurological function [63]. In adults, high intake of choline was associated with more efficient neural processing among overweight and obese individuals [64]. Furthermore, EFSA concluded a cause and effect relationship has been established between the consumption of choline and the maintenance of normal lipid metabolism, normal liver function, and normal homocysteine metabolism [65]. Phosphatidylcholine constitutes up to 40% of bile composition [66]. Free choline is re-absorbed within the small intestine [67], and only excess amounts are metabolized by the gut microbiome to produce TMA in the colon [32,39] (Figure 2). The single intake of up to 6 egg yolks (1 egg yolk equivalent to 119 mg choline) in 6 healthy volunteers resulted in high variability of post-prandial TMAO levels at 8 h (2–20 µM), returning to baseline values after 24 h [68]. In a study with 30 healthy males, fasting plasma TMAO concentrations were not affected after 4 weeks of ingestion of 400 mg/d choline, either via eggs (3 eggs/day) or choline supplementation [69]. Similarly, the intake of (phosphatidylcholine-rich) krill oil did not affect TMAO concentrations [70]. In a different study, consumption of up to 3 eggs/day for 4 weeks by 38 subjects beneficially increased high density lipoprotein cholesterol (HDL-c), reduced the low density lipoprotein cholesterol (LDL-c)/HDL-c ratio, and increased plasma choline, without changing fasting plasma TMAO concentrations [71]. Elevated circulating choline levels have been associated with CVD in some cross-sectional [72] and prospective studies [73]. The PREDIMED cohort reported positive associations between plasma choline and increased risk of major cardiovascular events [74]. In a recent cross-sectional study of 49 atrial fibrillation (AF) patients, plasma choline concentrations were higher than normal [75]. Results from 3 prospective cohort studies are also consistent with respect to the association between plasma choline and AF incidence [76]. A disruption of mitochondrial choline oxidation to betaine as part of mitochondrial dysfunction may precede the development of cardiovascular diseases. Oxidative stress and inflammation may impair betaine-homocysteinemethyltransferase activity, resulting in accumulation of circulating choline [77]. The relationship between circulating choline and T2D has been scarcely examined. Among diabetics a trend towards higher plasma choline was observed [78]; however, more recently, the PREDIMED study did not find any significant association [58].

Betaine is found in wheat bran, wheat germ, and spinach [60], and the usual intake is 100–400 mg/day [79]. Interestingly, increased plasma betaine levels were shown to be associated with lower risk of CVD [74]. However, recently, higher plasma concentrations of betaine were also associated with increased risk of AF [76]. Furthermore, two previous studies found that elevated plasma betaine concentrations were independently associated with HF risk [73,80]. Plasma betaine concentrations were reported to be reduced in insulin-resistant humans and were associated with insulin resistance [81] and a lower risk of T2D in a healthy Mediterranean population at high risk of developing CVD [58]. Even though betaine may be linked to TMAO production (Figure 2 and [37]), the single ingestion of 1760 mg betaine did not lead to an increase in urinary TMAO excretion in an experimental model [41]. Betaine appears to be a poor precursor for TMAO, as it has been shown to produce 100-fold less TMAO than choline [82].

Dimethylglycine is found in low levels in cereals, seeds, beans, and liver. Dimethylglycine is formed from betaine during the re-methylation of homocysteine to methionine by betaine-homocysteine methyltransferase (BHMT) [73] and is normally excreted in the urine or metabolized to sarcosine [83]. Dysregulation in dimethylglycine metabolism may lead to its accumulation in the plasma, which would inhibit BHMT activity, possibly causing an elevation in homocysteine concentrations [84]. Higher plasma concentrations of dimethylglycine were previously found to be associated with increased risk of acute myocardial infarction [85] and HF incidence in patients with CVD [73].

Ergothioneine is contained in mushrooms (bolete, oyster), kidney, liver, black beans, red beans, and oat bran [86]. It was found to have antioxidant and cytoprotectant properties [87] and was shown to be associated with a lower risk of cardiometabolic morbidity and mortality [88]. It is part of the biochemical network leading to TMA production, however its contribution to plasma TMA/TMAO levels requires further investigation [37].

Fish and seafood contain high amounts of TMAO, e.g., 311 mg per 100 g cod fillet [40]. TMAO has been shown to be directly absorbed by the intestine [37]. Similar to an acute supplementation of 2100 mg choline, 2970 mg L-carnitine, or 1670 mg TMAO by healthy volunteers, sea food consumption resulted in an increase in postprandial urinary excretion of TMA and TMAO after 24 h [41]. In a study with 40 healthy young men, consumption of meals producing TMAO provided directly from fish led to a significant increase in postprandial plasma TMA and TMAO levels, peaking at around 0.2 µM and 150 µM, respectively, 2 h post meals. However, egg or beef consumption, providing choline and L-carnitine, respectively, resulted only in negligible TMA/TMAO increases [40]. The difference between the studies could be due to different levels of L-carnitine and choline provided by supplementation or in the diet. TMAO has a plasma half-life of 4 h [40]. As is the case for plasma TMAO, urinary TMAO levels are also variable and observed in particular with high fish intake in humans [89]. In a human study with radiolabeled TMA or TMAO, 94.5% of the dose was excreted in the urine within 24 h, 4% of the dose was excreted in the faeces, and <1% in the breath [90]. In 20 Swedish men, high daily fish consumption elevated urinary TMA and TMAO levels to 47 (8.5–290) mmol TMAO/mol creatinine vs. 38 and 34 mmol TMAO/mol creatinine in non-fish consumers [91]. However, in most publications, only serum TMAO levels are considered as a marker.

Red meat contains both choline (138 mg in 100 g beefsteak) [92] and L-carnitine (49–144 mg in 100 g beefsteak) [93,94]. A diet high in red meat (258 mg L-carnitine and 573 mg choline per day), consumed for 4 weeks by 113 heathy omnivore volunteers, increased plasma TMAO levels from a baseline median level of 3.5 µM TMAO to 7 µM. Isotope tracing experiments revealed that TMAO was primarily generated from the L-carnitine; in addition, meat consumption led to a slight decrease in fractional renal TMAO excretion, possibly explaining in part the TMAO increase [95]. In a recent study, the Atkins diet, rich in red meat, increased plasma TMAO levels from 1.8 µM to 3.3 µM [96]. However, single consumption of 227 g of beef resulted in a 60-fold lower urinary TMA+TMAO excretion than 227 g of herring [41]. Furthermore, single consumption of 170 g of beefsteak only resulted in a negligible postprandial mean plasma TMAO increase, peaking 1 h after consumption [40].

## 3. Physiological Role/Effects of TMAO

In fish, TMAO functions as an osmolyte by supporting protein and nucleic acid stability by entropic and enthalpic mechanisms in the presence of high urea concentrations. This leads to maintaining cell volume under conditions of high osmotic urea and hydrostatic pressure stresses, which is especially important for the survival of the deep sea animals [97,98,99,100]. TMAO acts as a molecular chaperon and can also be used as such experimentally [101].

A mixture of osmolytes, including betaine and glycerophosphorylcholine, also counters the deleterious effects of highly concentrated urea and salt in mammalian kidneys [102]. If present at sufficient levels, TMAO and/or L-carnitine may support this process. L-carnitine (as well as betaine and erythritol) was also found to be an osmoprotectant and anti-inflammatory for human corneal epithelial cells [103].

### 3.1. TMAO Variability

Although the relationship between elevated TMAO and CVD/T2D is supported by several observational studies, TMAO concentrations were found to be highly variable. In a study including 100 healthy volunteers (median age: 63.0 years), plasma TMAO was measured at 2 time points one year apart. At both measurements TMAO varied within a range of 2–26 µM, although most values were within 2–8 µM. There was a greater within-person than between-person variability, making TMAO rather unsuitable as marker in epidemiological long-term studies [104]. A high plasma and urine TMAO variability was also reported in a study with overweight diabetic patients (median age 60 years), where TMAO concentrations as well as other osmolyte levels such as betaine were monitored four times at 6-month intervals over 2 years [105]. Thus, due to the observed intra-individual variability, increased TMAO may only serve as an indicator for associations with cardiometabolic diseases.

#### 3.1.1. TMAO Variability with Age and Sex

Older subjects were found to have high levels of TMAO. In a study comparing 168 young, 118 middle aged and 141 elderly adults, mean TMAO levels were around 2.5 µM, 4.8 µM and 10 µM, respectively. However, inter-individual variability was also observed within these populations [106]. Animal testing reported similar observations in old mice [107].

Gender differences may play a role in the observed TMAO levels and has been reported between men and women. Indeed, differences in TMAO levels are possibly due to differences in food consumption patterns levels [108]. However, the gender effect was not reported by others [29]. Animal models reported effects of gender on TMAO levels related to differences in the FMO expression. This is discussed in detail in the following paragraph.

#### 3.1.2. TMAO Variability through FMO

Some of the variability of TMAO in plasma levels was shown to be derived from inter-individual variability in FMO liver enzyme regulation [109]. FMO activity appears to vary with FMO isoform [110,111], as well as age [110,112]. Furthermore, menstrual cycle-dependent variations in TMA/TMAO occur in women, reflecting perhaps a hormonal-dependent FMO gene regulation, suggesting the possible function of TMA as a pheromone [34]. In addition, sex-specific differences in TMAO concentrations were reported in mice [110,113,114,115]. Thus, when both females and males are included, it is important to stratify the analysis according to sex, as female rats do have a significantly higher FMO_3_ expression.

In mice, FMO_3_ was identified as a central integrator of hepatic cholesterol and triacylglycerol metabolism, inflammation, and endoplasmic reticulum (ER) stress [116]. Knockdown of FMO_3_ in insulin-resistant mice prevented the development of hyperglycemia, hyperlipidemia and atherosclerosis [117], and reduced TMAO levels from 7 µM to 2.5 µM. A similar knockdown of FMO_3_ in LDL receptor knockout mice decreased TMAO levels only from 3.2 µM to 2.2 µM, while still affecting lipid and glucose metabolism [118]. This suggests that FMO_3_ may have a role in metabolic pathways and disease progression independently from TMAO [118]. Furthermore, it has been reported that FMO enzymes are involved in steroid hormone synthesis, important in metabolic disease etiology [119].

In a different study, FMO_3_-deletion protected choline supplemented LDL-receptor-/- mice from obesity, while reducing TMAO levels from 2.5 µM to 0.4 µM [120]. Interestingly, in various different FMO deficient mice (fmo1-/-, 2-/-, 4-/-, 5-/-), there was no correlation between atherosclerosis and either TMAO production or urinary TMAO concentration. The plasma total cholesterol concentration was negatively, but weakly, correlated with TMAO levels [110].

Variability in FMO_3_ expression was reported in patients with CKD, leading possibly to a variability in renal clearance of TMAO [121]. These effects could be the results of activation in FMOs elicited by octylamine and uremic serum [122]. Lower clearance in TMAO coincided with increased FMO_3_ expression in a CKD mouse model [123]. High plasma TMAO levels in CKD patients may contribute to the observed damage in blood vessels [122]. In pediatric CKD patients, plasma TMAO, DMA, and TMA levels were also increased, while the level of these compounds was decreased in the urine [124]. The microbiome was found to be altered in children with CKD where Cyanobacteria, Subdoligranulum, Faecalibacterium, Ruminococcus, and Akkermansia abundance were affected [124]. In a different study with CKD children, the Prevotella genus was reduced, while Lactobacillus and Bifidobacterium were increased [125,126]. More recently, Rath and colleagues reported the role of diet and age in the abundance of the TMA-producing bacteria and how diet and supplementation can modulate the abundance of the microbiota leading to the formation of TMA [127]. In particular, bacteria from Clostridium XIVa and Eubacterium sp., containing choline TMA-lyase genes, and Gammaproteobacteria, containing L-carnitine oxygenase genes, have been identified in the gut microbiota [128].

#### 3.1.3. TMAO Variability by Diet

The influence of diet on TMAO levels has been examined in several studies. A diet with lower carbohydrate and high resistant starch content was associated with slightly higher plasma TMAO levels [129]. Interestingly, a paleolithic diet rich in whole grains also led to lower serum TMAO and abundance of TMA producer Hungatella [130]. In 115 healthy people at increased risk of developing colon cancer, a Mediterranean (moderate in fish) diet did not cause significant changes in the (fasting) TMAO concentrations after 6 months of supplementation [131]. Daily red meat consumption (providing 258 mg L-carnitine/day) [95] caused elevations in TMAO median levels of around 7 µM (interquartile range 4–11 µM). However, non-red meat, egg, or fish consumption in 271 subjects was not associated with elevated fasting TMAO, choline, or betaine concentrations, despite a slightly positive association between dairy food consumption and plasma TMAO concentrations [132]. Furthermore, in a 2-year study with 100 healthy volunteers, there were no associations between TMAO and the consumption of animal food (low fish-intake population); however, slightly higher TMAO levels (and lower levels of betaine and choline) were observed in prebiotics consumers [104].

Depending on the precursors, different chemical intermediates may be formed by the gut microbiome similarly to TMA (Figure 2). For instance, L-carnitine was found to be converted to gamma-butyrobetaine by several commensal bacteria; gamma-butyrobetaine in turn was found to be converted to TMA by a strain of the Clostridiales. The second conversion step took place at a low rate in omnivorous humans (*n* = 40) and at a very low rate in vegans/vegetarians (*n* = 32) [133], perhaps linking diet to microbiota differences and TMA formation. Previously, similar results were published showing that following a 250 mg L-carnitine challenge, omnivores had significantly higher plasma TMAO concentration, peaking at 14 µM 8 h post ingestion (*n* = 5) compared to vegans/vegetarians, slowly ascending to 1 µM 24 h after ingestion (*n* = 5) [94]. However, when larger populations are studied, fasting TMAO levels were about 2 µM in vegans/vegetarians (*n* = 26) and almost 3 µM in omnivores (*n* = 51) with the upper 95 percentiles of 4.5 µM in vegans/vegetarians and 7 µM in omnivores [94]. It is believed that vegetarians/vegans adapted to low-carnitine diets have a higher L-carnitine bioavailability than carnivores, because dietary L-carnitine is more efficiently absorbed [134] and therefore less available to the microbiome.

Altogether, dietary precursors would have short-term influence on postprandial TMAO levels but would not lead to lasting elevated fasting TMAO levels.

#### 3.1.4. TMAO Variability by Microbiome Variability

As discussed, the microbiome is the central determinant for TMAO generation [135]. The TMAO precursor, TMA, is generated microbiologically within the intestine, and antibiotics were shown to suppress TMAO formation [94,115,136]. In human subjects (*n* = 5) ingestion of 250 mg radiolabeled L-carnitine together with a 227 g steak (providing an additional 180 mg L-carnitine [94]), resulted only in a modest TMAO increase in the plasma (2.5 µM TMAO 24 h post ingestion) and urine. If the L-carnitine challenge was repeated following microbial suppression using antibiotics, no TMAO could be detected. Three weeks later, the same L-carnitine challenge produced 12 µM labelled TMAO at 24 h from a (possibly still dysbiotic [137]) repopulated microbiome [94]. Gnotobiotic mouse studies showed that TMAO accumulates in the serum of mice colonized with TMA-producing species [138]. These results suggest that diet and in particular supplementation with TMA precursors can modulate the microbiome toward proliferation of TMA-producing bacteria.

In 40 healthy young men, the dietary precursors choline from eggs and L-carnitine from beef were converted to TMAO more efficiently in individuals with a higher Firmicutes to Bacteroidetes ratio and less gut microbiota diversity, demonstrating that TMAO production is a function of individual differences in the gut microbiome. Interestingly, baseline TMAO concentrations were 3.4 µM for both low (*n* = 15) and high (*n* = 11) TMAO producers [40], pointing again to individual differences in the microbiota.

An oral L-carnitine challenge test was designed to identify individuals with a TMAO-producer phenotype of their gut microbiota [139]. This also puts emphasis on the microbial composition of the microbiota, relevant for TMAO production and levels.

In a more pronounced and harmful way dysbiosis can also alter the microbiome. Dysbiosis is part of the disease network including cardiometabolic diseases (Figure 1). Dysbiosis in turn may alter TMAO levels, by either decreasing or increasing TMA producing strains within the microbiome. In some cases, TMA-producing bacteria are somehow reduced by dysbiosis [140,141,142]. In other cases, it has been shown that increased TMAO correlates with a dysbiotic microbiome [143,144]. Dysbiosis is known to be triggered by factors like an unhealthy diet, especially a high-animal fat diet [12]. However, several different factors can contribute to dysbiosis including the cardiometabolic disease network (Figure 1). Furthermore, dysbiosis contributes to the progression of CVDs by promoting major CVD risk factors: atherosclerosis and hypertension [12]. Dysbiosis has been shown to promote kidney disease, since the intestinal barrier becomes more permeable for microbially generated metabolites [145,146,147]. Relative abundance of Akkermansiamucinophilia in colon biopsies was shown to be inversely correlated with TMAO levels [131]. A. mucinophilia is a mucotroph bacterium and an indicator of a functioning intestinal mucus layer [126,148].

Altogether, the amount of microbial TMA formation largely depends on the composition of the intestinal flora as well as the state of the mucus layer, both of which can be influenced by several factors, including but not limited to diet and disease status (Figure 1).

Increases in TMAO levels resulting from TMA increases are plausible but provide no statement regarding any effects on health or disease by TMAO.

So far, only associations from observational studies have been described. In order to distinguish cause and effects, intervention and mechanistic studies are needed.

## 4. TMAO and Cardiometabolic Diseases

### 4.1. TMAO Levels in Humans Associated with CVD and T2D

Due to the associations of elevated TMAO levels with cardiovascular [136] or metabolic outcomes in diseased populations, TMAO has been used as a disease biomarker [28,38,149]. In this section the information is also summarized as a table (Table 1).

In a cross-sectional study, elevated plasma TMAO levels were found in 50 subjects with cardiovascular complications (heart attack, myocardial infarction, MI, stroke, or death), as compared to subjects (*n* = 50) without complications [115]. These results were confirmed in a cohort of 1876 subjects [115]. In 2595 subjects undergoing cardiac evaluation, plasma TMAO levels above the median of 4.6 µM predicted an increased CVD risk; elevated plasma dimethylglycine levels, a metabolite of choline, rose modestly but significantly from 4.29 to 4.5 µM with increased CVD risk prediction [94]. Within a population of 3903 subjects undergoing elective diagnostic coronary angiography, the median TMAO level was 3.7 µM (interquartile ranges 2.4–6.2 µM). High plasma choline and betaine levels were associated with higher risk of mortality, MI, or stroke with concomitant increase in TMAO in 3903 sequential stable subjects undergoing elective diagnostic coronary angiography [82]. In a prospective study of 622 hypertensive stroke patients and 622 matched controls, the serum TMAO level in the highest tertile was ≥3.19 μM/L, while in the lowest tertile it was <1.79 μM; the high TMAO levels were associated with increased risk of stroke [160].

A recent systematic review and meta-analysis of 19 prospective studies (*n* = 19,256, including 3315 incident CVD cases) concluded that elevated TMAO concentrations were associated with a 62% increased risk for all-cause mortality. However, individuals with elevated concentrations of TMAO precursors (L-carnitine, choline, or betaine) had only 1.3 to 1.4 times higher risk for major adverse cardiovascular disease events compared to those with low concentrations [169].

Another meta-analysis of 11 prospective studies with a total of 10,245 patients found that elevated circulating TMAO was associated with a 23% higher risk of cardiovascular events and 55% higher risk of all-cause mortality. Mean elevated TMAO levels ranged from 4.1 µM to 72.2 µM. Control subjects with the lowest risk had levels ranging from 2.3 to 32.2 µM [170]. When studies with missing control values and unusually high control values (27.5 and 32.2 µM) were omitted, TMAO levels [80,136,153,158] in the remaining 5799 patients ranged between 4.1 µM and 12.0 µM (mean 6.8 µM) for high-risk patients, and between 2.3 µM and 2.9 µM (mean 2.6 µM) for low-risk patients.

Recently, a prospective study investigated the relationship between 10-year changes in plasma TMAO levels and coronary heart disease incidence among 760 healthy women. Increases in TMAO levels during the follow-up were associated with higher coronary heart disease risk, and this relationship was strengthened by unhealthy dietary patterns and attenuated by healthy dietary patterns [171]. This points again to an effect of the microbiome independent from TMAO metabolism.

In a study with 435 patients undergoing elective cardiac risk factor evaluation, plasma TMAO concentrations were higher in subjects with T2D (around 4.9 µM) compared to subjects without T2D (3.3 µM TMAO), with high variability, however, between subjects [120]. In 353 CAD patients, increased plasma TMAO levels (median 5.5 µM; interquartile range 3.4 to 9.8 µM) were associated with a higher SYNTAX score (synergy between percutaneous coronary intervention with taxus and cardiac surgery; a measure of disease severity), but TMAO levels were not associated with subclinical myonecrosis [154].

In 283 subjects with a mean age of 66.7 years, T2D was associated with a higher plasma TMAO concentration (8.6 ±12.2 µM compared with 5.4 ±5.2 μM) [152]. In a case-control study with a cross-sectional design, a positive association between plasma TMAO concentrations and gestational diabetes mellitus was observed [172], where an inverse association was found in a recent prospective cohort [173]. Both studies were conducted among Chinese pregnant women. In a cross-sectional study with 137 adults, serum TMAO levels increased along with body mass index, visceral adiposity index, and fatty liver index; serum TMAO ≥8.74 µM was considered as predictive of metabolic syndrome [28]. In a recent prospective study with 300 T2D-free adults, a nonlinear association between baseline plasma TMAO and prevalent prediabetes was reported [174].

In 859 venous thromboembolism patients, TMAO levels showed a U-shaped association with mortality, with optimum levels around 4 µM [162]; however, TMAO was not associated with bleeding events in 81 patients with stable angina [175].

In a cross-sectional study with 271 healthy adults, plasma TMAO concentrations were positively associated with inflammation—participants in the top TMAO quartile (3.13 μM) had higher plasma concentrations of TNF-α, sTNF-R p55, and sTNF-R p75 than participants in the bottom quartile (2.08 μM) [132]. Converse results were found in a case-cohort study with development of 251 incident T2D cases in a random sample of 694 participants within the PREDIMED trial. These subjects were healthy and T2D free at baseline. After adjustment for recognized T2D risk factors, individuals with elevated plasma TMAO levels at baseline had a lower risk of T2D [58].

In another recent cross-sectional study using data collected from 1653 adults, higher plasma TMAO concentrations were associated with an adverse cardiometabolic risk profile and with a number of TMA-producing bacterial taxa (associations of plasma trimethylamine N-oxide, choline, carnitine, and betaine with inflammatory and cardiometabolic risk biomarkers and the fecal microbiome in the Multiethnic Cohort Adiposity Phenotype Study.) [176].

A recent study suggested an association between TMA rather than TMAO and cardiovascular diseases. In patients with severe aortic stenosis, TMAO levels were 5.5 µM (controls 3.6 µM); at the same time, TMA levels were 59.5 µM (controls 23.2 µM) [161]. TMA but not TMAO was also found to be associated with increased risk of gestational diabetes [173].

Altogether, one can conclude from the above literature that discrepancies are reported from different observational studies.

TMAO elevation may be just a feedback mechanism to counter disease progression in light of its role as a molecular chaperone supporting proper protein folding (see above) and other potential beneficial roles [177], as will be discussed later in this review. In addition, the microbiome involved in the metabolism of TMAO can play an independent role in disease progression, making TMAO a marker and not an inducer of disease.

### 4.2. TMAO in Patients with Compromised Renal Function

TMAO and TMA are known as microbiota-generated molecules, along with p-cresyl sulfate, indoxyl sulfate, and indole-3 acetic acid [161,178,179]. Absorbed from the intestine into the blood stream or metabolized by the liver, TMA and TMAO may accumulate if not excreted properly in patients with compromised renal function.

Along this line, a recent review suggested that elevated TMAO in the circulation could reflect T2D or atherosclerosis mediated renal dysfunction, which reduces TMAO excretion, thereby elevating TMAO plasma concentrations [8]. This suggests that TMAO may be a marker of disease, but not a causative factor.

It has been reported that diabetic CKD patients (*n* = 20) had an increase in TMA-producing bacteria leading to elevated plasma TMAO levels (1.516 µg/mL = 20.2 µM) compared to control subjects (0.183 µg/mL = 2.4 µM) [167]. In another study with 521 CKD patients, the median TMAO level was 7.9 µM (interquartile range, 5.2–12.4 μM), compared to a median of 3.4 µM TMAO in 3166 control subjects. In hemodialysis patients (with more severely impaired kidney function) TMAO levels reached 77 ± 26 µM, compared to 2 ± 1 µM in control subjects [163].

More studies are needed to fully understand the TMAO variability in this disease. It is possible that individual variability or disease severity in these patients cause variable TMAO levels. This is supported by the following observation: only a few patients with severe renal disorders (hemodialysis patients [163]; see Figure 1) have extremely high TMAO levels.

### 4.3. Cause to Effects

A recent bi-directional Mendelian randomization analysis examined the causal direction between gut microbiota-dependent TMAO, or its precursors, and cardiometabolic diseases such as risk of T2D, CAD, MI, stroke, AF, and CKD. Genome-wide association with TMAO, L-carnitine, choline, and betaine levels, and individual single nucleotide polymorphisms (SNP), allowed to the nature of these associations to be identified. Genetically predicted higher TMAO and L-carnitine were not associated with higher odds of T2D, AF, CAD, MI, stroke, and CKD after Bonferroni correction for multiple comparisons (*p* < 0.0005). However, higher choline was associated with an increased risk of T2D while higher betaine was associated with a decreased risk. In addition, it was reported that T2D and CKD were causally and directionally associated with higher TMAO levels, suggesting that these diseases are the cause of plasma TMAO elevation and supporting the feedback mechanism of TMAO increase as a result of disease [180]. This is supported by preventive animal studies described below.

For kidney disease, the lower excretion of TMAO by affected patients explains higher TMAO levels. In addition, CKD progression goes along with an increased gut permeability, which in turn sets free an overload of microbiota-generated compounds, including TMAO [178,181].

Altogether, the potential role of TMAO in cardiometabolic diseases, as previously proposed [94,115,169,170], requires further evaluation—as done below.

## 5. TMAO in Experimental Models

### 5.1. TMAO in Cell Culture

Controversial data have been reported in cell culture experiments. With high TMAO concentrations, inflammatory signals and cellular damage were observed. Human aortic endothelial cells had proinflammatory responses to extremely high levels of TMAO (200 µM), including the induction of COX-2, E-Sel, ICAM-1, and IL-6 and activation of mitogen-activated protein kinase and nuclear factor-κB (NF-κB) pathways [182].

In human platelets, the most pronounced changes in thrombin occurred with 100 µM TMAO, lower concentrations were less effective [183]. In cultured carotid artery endothelial cells, 30 µM TMAO increased caspase 1 activity, enhanced IL-1β production, and led to inflammasome formation [184]. Endothelial progenitor cells cultured with 200 and 500 µM TMAO promoted cellular inflammation, oxidative stress, and suppressed cellular functions, whereas 2 µM or 100 µM TMAO did not have such effects [175]. In a dose-response experiment on macrophages, TMAO up to 100 µM had no effect on inducing foam cell formation, the first steps in atherosclerosis development [185]. Furthermore, in isolated rat cardiomyocytes, 10 or 100 µM TMAO applied for 1 to 24 h did not affect cell viability, sarcomere length, intracellular reactive oxygen species (ROS), and mitochondrial membrane potential. Furthermore, the simultaneous treatment with TMAO and known cardiac insults, such as H_2_O_2_ or doxorubicin, did not enhance the treatment’s effects [186].

A recent study evaluated the effects of TMA and TMAO on rat cardiomyocytes. TMA at 1000 µM showed more adverse effects than TMAO at 100.000 µM. The concomitant treatment with TMA (1000 µM) and TMAO (100.000 µM) protected cardiomyocytes against the deleterious effects (cell shrinkage, detachment) of TMA [161]. These experiments have to be assessed with caution as all unfavorable effects were observed with TMAO concentrations well beyond the concentration range observed in humans affected by CVD, T2D, or other aspects of the disease network. It is unclear how TMAO mediates these effects. Recently, TMAO was found to bind to endoplasmic reticulum stress kinase PERK (EIF2AK3), selectively activating the PERK branch of the unfolded protein response. This induced the activation of the transcription factor FoxO1, which has been described to suppress lipogenesis [187].

Currently, reliable proof for any effects at physiological or slightly elevated levels (e.g., up to 12 µM) of TMAO as a direct cause or as a risk factor of cardiac damage at the cellular level is missing. On the contrary, TMAO may have beneficial effects, as described above.

### 5.2. TMAO in Animal Studies

The pieces of evidence for effects of TMAO or its precursors in animal experiments are summarized in Table 2 (unfavorable effects) and Table 3 (neutral or beneficial effects).

However, several factors including experimental design (intervention vs. prevention), diet, gender, and non-physiological high concentrations have led to contradictory results, similar to the culture experiments above. In addition, not all test systems are equally well suited to examine a possible role of TMAO in disease formation or prevention: several animal models may increase the frequency of disease symptoms by mutations or other distress to the animals in order to increase the occurrence of measurable events.

The majority of data were collected in the apoE−/− mouse model (apolipoprotein E-knockout). ApoE is a multifunctional protein with central roles in lipid metabolism; it transports lipids, including cholesterol, through the cerebrospinal fluid and plasma. Unlike humans, mice lack the cholesteryl ester transfer protein (CETP), which also transfers cholesterol ester from HDL-c to LDL-c particles. Normal mice have the majority of circulating cholesterol being carried by HDL particles, so they never develop atherosclerotic lesions on a chow diet; this is changed in the apoE−/− mouse model, where the ApoE protein is inactivated, which leads to enrichment in very low-density cholesterol (VLDL-c) [210]. Thus apoE−/− mice develop atherosclerotic lesions on a chow (plus fat) diet due to the resulting pro-atherogenic lipoprotein profile. Lesion formation collected in apoE−/− mice was previously considered to originate from increased oxidative stress [114]. Due the lack of CETP and other human-mouse differences, results obtained from apoE−/− should be interpreted carefully. A mouse strain with transgenes for two dyslipidemia-inducing mutations, the human apolipoprotein E-Leiden (APOE-Leiden), and human cholesteryl ester transfer protein (CETP) [211] may be more close to the human situation. Interestingly, when the E-Leiden/CETP mouse model is used, TMAO showed protective effects in preventing atherosclerosis development [185].

In a study of germ free apoE−/− mice on a high-fat Western-type diet, atherosclerosis developed, with no measurable difference to animals raised conventionally (with germs) [212]. However, if a similar experiment was done in germ free apoE−/− mice on regular chow, differences in atherosclerotic plaques between germ-free and conventionally reared apoE−/− mice became apparent. The latter group had fewer plaques due to a protective effect of the microbiota [213].

In some studies, a structural analogue of choline, 3,3-dimethyl-1-butanol (DMB), was used. DMB inhibits TMA formation from cultured microbes or physiologic polymicrobial cultures as well as distinct microbial TMA lyases concurrently with inhibition of foam cell formation and atherosclerotic lesion development. This lead to the conclusion that DMB can be used as a treatment for atherosclerosis [214]. In addition, DMB was shown to prevent hypertension in experimental animals [215].

#### 5.2.1. Unfavorable Effects of TMAO in Animals

In several nonclinical settings (see Table 2), high TMAO levels (25 to 400 µM) were associated with CVD symptoms, e.g., atherosclerotic plaques. Control groups had fewer CVD symptoms and lower TMAO levels. However, in most of these studies, experimental animals were severely affected by disease, old age, and/or other factors. For instance, animals with partially ligated carotid artery/coronary ligation [184,191,193] are ill, in pain, and stressed from the operation; this procedure may severely perturb the system and induce dysbiosis. Interestingly, severe cardiovascular stress caused by coronary ligation increased TMAO levels up to 30 µM in rats; however, sham operations also caused slightly higher than baseline TMAO levels (10 µM), likely due to postoperative dysbiosis [193].

In most animal experiments, high TMAO (0.12–0.3% of chow/drinking water) or L-carnitine or choline supplementation (≥1%) resulted in plasma levels between 30 and 100 µM [94,115,165,182,191,193], in some cases up to 400 µM TMAO [94,192], and many of these studies were conducted in female mice, which express high FMO3 and give rise to high TMAO concentrations in plasma. It should be noted that 1.3% L-carnitine in the diet of a mouse approximately corresponds to 2080 mg L-carnitine/kg (see calculation in the footnote of Table 2), which is an excessive dose, corresponding to a human equivalent dose of 11,818 mg L-carnitine/day [185]. Acute high TMAO (achieving 91 µM TMAO) injection experiments increased thrombus formation in a carotid artery injury mouse model [183]. TMAO injection in LDL-receptor deficient mice (86 µM plasma concentration) induced increased levels of inflammatory markers and activated the mitogen-activated protein kinase, extracellular signal-related kinase, and nuclear factor-κB signaling cascade [182]. The relevance of these results for healthy humans, or even those affected by CVD or metabolic syndrome, remain limited.

In apoE−/− mice, chronic dietary L-carnitine supplementation (1.3%), corresponding to 2080 mg L-carnitine/kg, altered microbial composition and increased aortic root atherosclerotic lesions by 1.8-fold, while enhancing TMA/TMAO synthesis. However, this did not occur if intestinal microbiota were suppressed by antibiotics [94]. Similar effects (including atherosclerotic inflammation and foam cell formation) in the same mouse model were observed with a 1.0% choline supplementation [115]. Although significant, the difference in aortic lesion between supplemented vs. control was observed in only three mice, suggesting a possible type I error.

In a mouse study, high choline or TMAO supplementation (corresponding to 1600 mg choline/kg or 192 mg TMAO/kg) resulted in renal damage [165]. This study may explain a vicious cycle effect of further renal damage, once TMAO levels are extremely and unusually high.

#### 5.2.2. Favorable or Neutral Effects of TMAO in Animals

When TMAO or its precursors are supplied in lower and recommended doses, protective effects on atherosclerosis hypertension and cholesterol are observed (Table 3), [185,202,206,208]. Protection against atherosclerosis was observed in mice supplemented with 352 mg/kg L-carnitine, corresponding to 2000 mg per day in humans. The resulting plasma TMAO levels were significantly elevated compared to baseline [185].

Moderate TMAO supplementation (approximately 6.7 mg TMAO/kg) also increased plasma TMAO levels by four- to five-fold while reducing diastolic dysfunction in the pressure-overloaded hearts of hypertensive rats [206].

Importantly, adverse effects on atherosclerosis or inflammation could not be observed in any of the studies detailed in Table 3, even where supplementation was relatively high, suggesting that other factors in the experimental design may interfere in the reported effects [207,209].

It should be noted that the animal models included in Table 3 are not severely compromised (e.g., impaired kidney function), as reported in Table 2, further emphasizing the difference in response to TMAO between relatively healthy and diseased populations. It is also possible that the adverse effects reported in Table 2 at levels up to 400 µM reflect non-specific effects.

#### 5.2.3. Why TMAO Is a Marker but Not the Effector

As explained above, one of the dietary precursors of TMAO generation is L-carnitine. Furthermore, TMAO can become bioavailable as a result of consumption of fish and other sea foods (Figure 2). Evidence on the beneficial effects of L-carnitine and TMAO on human health is summarized below.

A recent review on L-carnitine and heart disease emphasized the role of exogenous L-carnitine as a preventive and/or therapeutic strategy against CVD, as well as associated dysfunctions including hypertension, hyperlipidemia, diabetic ketoacidosis, hyperglycemia, type 1 diabetes, insulin resistance, and obesity [52].

A systematic review and meta-analysis of 13 controlled trials with 3629 patients experiencing an acute myocardial infarction demonstrated that L-carnitine was associated with a 27% reduction in all-cause mortality, a 65% reduction in ventricular arrhythmias, and a 40% reduction in angina symptoms [216]. Although a more recent meta-analysis of 17 randomized controlled trials did not find significant differences in all-cause mortality with 1625 chronic HF patients compared to controls, L-carnitine was effective in improving clinical symptoms and cardiac functions and decreasing serum levels of heart insufficiency marker proteins [217]. Another meta-analysis of 5 studies revealed that L-carnitine was useful in treating patients with insulin resistance [218].

The following studies supplying L-carnitine (or a derivative) also evaluated TMAO levels in addition to cardiovascular effects.

In healthy, aged women, 24 weeks of 1500 mg/day L-carnitine Tartrate supplementation (*n* = 11) significantly increased plasma TMAO concentration (35 µM after 24 weeks), a 4.4-fold increase compared to the placebo group (*n* = 9; 8 µM) with no lipid profile changes or other markers of adverse cardiovascular events [47]. The lack of change in disease markers demonstrates that even a moderate elevation of TMAO in otherwise healthy individuals will not cause cardiometabolic dysfunctions. Furthermore, TMAO was shown to be an independent risk factor for disease and induced atherosclerosis in animal models without changing lipid profiles, so some may argue that a lack of change in lipid profiles and other markers in the face of elevations of TMAO is not necessarily indicative of a lack of cardiometabolic dysfunctions.

In 31 hemodialysis patients, 900 mg/day L-carnitine supplementation for 6 months increased total, free, and acyl L-carnitine, as well as plasma TMA and TMAO (from 222.5 µM to 548.4 µM, a factor 2.5; healthy control 174.3 µM—possibly method-derived high baseline values only allow a relative comparison). The supplementation decreased markers of vascular injury and oxidative stress such as intracellular adhesion molecule-1, vascular cell adhesion molecule-1, and malondialdehyde levels, thereby reducing advanced glycation end products and suggesting beneficial effects in these patients [48]. Therefore, there is accumulating evidence for beneficial cardiovascular effects of L-carnitine supplementation or a L-carnitine-rich diet.

Considering the above stated correlation between TMAO levels and cardiometabolic outcomes, a controversy also arises from the fact that the consumption of fish and seafood was previously shown to be associated with significant reduction in the risk of coronary heart disease-associated mortality [219]. In addition, 1–2 servings of fish per week were found to reduce the risk of coronary heart disease death by 36% and total mortality by 17% [220]. In 2 population-based cohort studies with 134,296 men and women, fish intake was inversely associated with risk of total and cause-specific mortality [221]. In a randomized controlled trial, a lean-seafood diet—compared to a non-seafood control—significantly improved fasting and the postprandial lipid profile [222]. Evidence on the potential effects of fish consumption on lipids arising from intervention trials indicates that consuming oily fish led to significant improvements in triglycerides and HDL-c levels [223].

With a Mediterranean diet (moderate amounts of fish), CVD incidence could be significantly reduced compared to a control diet [224]. Data did not support a significant association between basal plasma TMAO levels and increased risk of CVD in a Mediterranean population at high cardiovascular risk (PREDIMED-study) [74]. Interestingly, the PREDIMED study documented an inverse association between high plasma TMAO levels at baseline and incidence of T2D [58].

## 6. Summary and Conclusions

Given the complexity of cardiometabolic diseases and based on (1) the fact that observational trials cannot resolve the cause from the effects, (2) the recent PREDIMED data [58,74] measuring TMAO at baseline, and (3) the recent bi-directional Mendelian randomization analysis [180] examining the causal direction between TMAO and its precursors and the diseases, it can be concluded that TMAO increases in patients are the result (rather than the cause) of disease.

The apparent contradictions and the results from animal studies have to be interpreted in light of several factors. Animal studies, which apply excess amounts of TMAO or its precursors in distress-suffering animal models (e.g., coronary ligation models), are far from the normal situation and cannot explain a causal relationship between TMAO and disease development.

On the other hand, several animal experiments using lower doses of TMAO and its precursors did show protective effects.

The most convincing data in humans are derived from studies on fish consumption or intake of the dietary supplement L-carnitine. Indeed, the increased plasma TMAO levels observed in CVD/T2D may be a compensatory mechanism. Furthermore, the type of precursor for TMAO generation may also be important.

Increases in TMAO levels during CVD/T2D progression (beyond the observed wide natural intraindividual variability in TMAO levels) could be the result of disease-related dysbiosis. Another factor in such variability is related to the methodology itself. Indeed, most studies reported plasma TMAO only without considering TMA or urinary secretion. Perhaps it would be more relevant to consider the plasma TMA/TMAO ratio as a marker. It is also possible that TMAO elevation is a result of injuries caused by disease (e.g., atheroma lesions), which will induce TMAO up-regulation to promote the healing process. TMAO has been shown to be up-regulated in ischemic-injury models, as discussed above. These up-regulations may be mediated by FMO gene regulation, which in turn is modulated by many factors.

In conclusion, increased TMAO levels may be a compensatory mechanism in response to disease. Further research and intervention studies with relatively low levels of TMAO may be needed to confirm this theory.

## 7. Literature Search Methods

The PubMed database (www.ncbi.nlm.nih.gov) was searched for the term “Trimethylamine N-oxide” or “TMAO”. Of the total 1963 references initially identified from the literature search up to 11 November 2019, 179 hits and 67 hits were filtered out for “review” and “clinical trial”, respectively, as well as 334 hits for “rat or mouse” (Figure 3). Hits were screened for relevant publications, both for giving a literature overview and for finding nonclinical and clinical studies evaluating the hypotheses that TMAO is a cause or effect of cardiometabolic outcomes. Literature cited within articles was followed up to complement the search. Additional literature sources were added during the review phase up to 27 April 2020.

## Figures and Tables

**Figure 1 nutrients-12-01330-f001:**
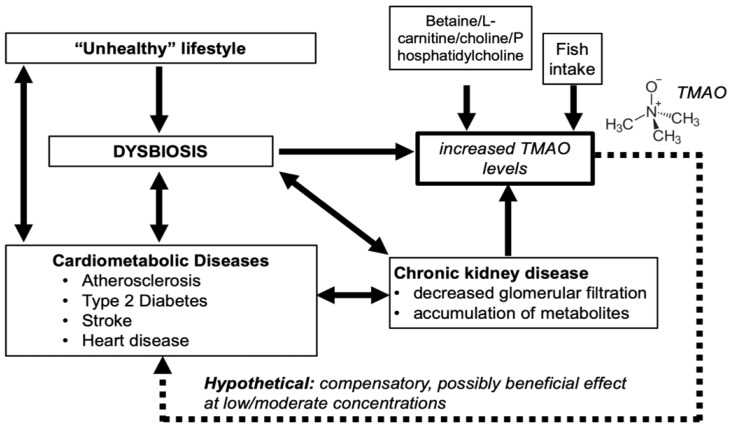
Disease network around cardiovascular disease and metabolic syndrome. Trimethylamine N-oxide (TMAO) can originate either directly from fish consumption or indirectly from intake of dietary precursors (e.g., L-carnitine, choline, or betaine). Dysbiosis has a major influence on cardiometabolic diseases or disease factors. During cardiovascular disease/type 2 diabetes, TMAO levels were found to increase to 4–12 µM in patients, possibly resulting from a disturbed microbiome and/or a decreased intestinal barrier. In the kidney, TMAO is rapidly excreted via the urine. Increased TMAO levels, as observed in studies with patients, may signal a decreased renal function. Unfavorable contribution of TMAO may take place at extremely high concentrations in patients with severely impaired renal function, e.g., hemodialysis patients (dotted arrow). Moderately increased TMAO levels may be a compensatory mechanism in diseased populations. Arrows pointing in two directions: both affect each other.

**Figure 2 nutrients-12-01330-f002:**
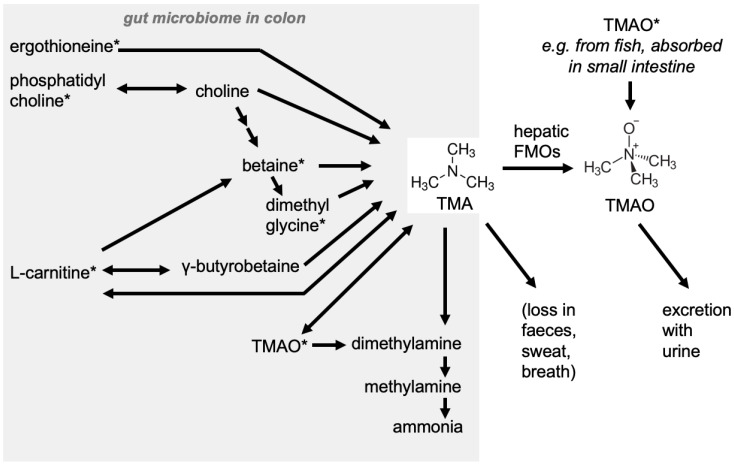
Main biochemical conversions leading to trimethylamine (TMA) and TMAO. Modified from [37,38]. The compounds with an asterisk can originate from the diet. The conversions within the grey box (gut microbiome) are induced by the microbiota (mainly Firmicutes, Proteobacteria), and the resulting TMA is absorbed within the colon and converted to TMAO by liver FMOs (flavin monooxygenases). The possibly partial microbial degradation of TMA and TMAO (by methylotrophs and other occasional bacteria (Pseudomonas/Bacillus)) can result in the formation of formaldehyde within the colon; also, methylamines (including TMA/TMAO) could be substrates for the formation of nitrosamines [32,39]. If TMA is absorbed from the colon, 95% of it is converted to TMAO, which is then excreted in the urine [34]. Arrows pointing in two directions: reactions go in both directions.

**Figure 3 nutrients-12-01330-f003:**
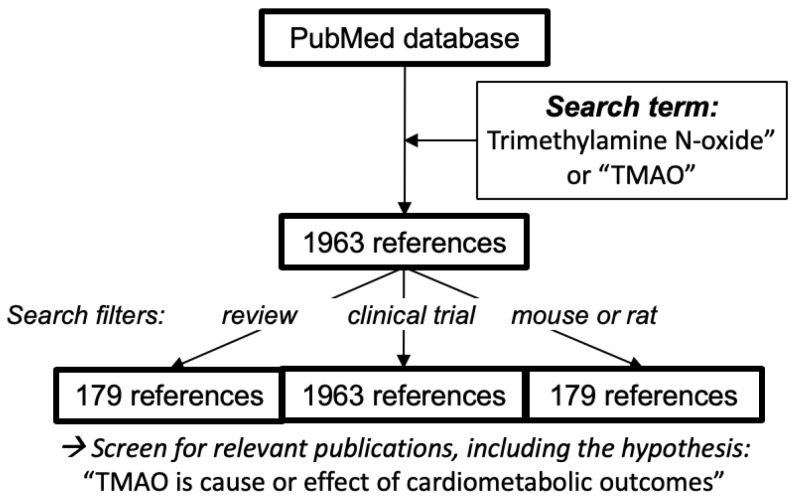
Search method for finding literature on TMAO. A total of 1963 references (up to November 2019) on TMAO were further searched with the indicated search terms. Additional references were found by following up citations within the literature and during the review phase in April 2020.

**Table 1 nutrients-12-01330-t001:** TMAO levels in patients with cardiovascular diseases and renal dysfunction.

First Author, Year and Citation	Number of Subjects	Study Population	Mean Age (Years)	TMAO Level Median (or Mean) (µM)	TMAO Interquartile Range (µM)	Increased Disease Severity with Higher TMAO Levels
Tang, 2013 [136]	720	Patients with stable heart failure undergoing cardiac evaluation	63.0	5.0	3.0–8.5	yes
Koeth 2013 [94]	2595	Patients undergoing elective cardiac evaluation, GeneBank study	62	4.6	--	yes
Bae 2014 [150]	835	Postmenopausal women with colorectal cancer within the Women’s Health Initiative Observational Study	66	4.0	2.9–6.0	yes
	835	Postmenopausal women healthy controls	67	3.8	2.6–5.7	NA
Lever 2014 [80]	79	Coronary Disease Cohort Study (CDCS), participants with T2D	74.0	7.5	4.4–12.1	yes
	396	CDCS participants without T2D	68.0	4.8	3.0–9.1	yes
Wang 2014 [82]	3903	Patients undergoing elective diagnostic coronary angiography	63	3.7	2.4–6.2	yes
Tang 2015 [151]	112	Patients with chronic systolic HF	57.0	5.8	3.6–12.1	yes
Obeid 2016 [152]	283	Subjects participating in a diabetes case-control study or a vitamin-supplementation trial	66.7	4.36 **	Not indicated	yes
Ottiger 2016 [153]	317	Community-acquired pneumonia patients	72.0	3.0	1.7–5.4	Yes
Rohrmann 2016 [132]	104	Healthy men	50	(geometric mean 2.55	(95% CI 2.17–2.99)	Only TNFα but not CRP and IL-6
	167	Healthy women	44	(geometric mean 2.52)	(95% CI 2.22–2.86)	
Senthong 2016 [154]	353	Atherosclerotic CAD patients	65.0	5.5	3.4–9.8	yes
Senthong 2016 [155]	2235	Patients with stable CAD who underwent elective coronary angiography	63.0	3.8	2.5–6.5	yes
Senthong 2016 [156]	821	Patients with peripheral artery disease	66	4.8	2.9–8	yes
Suzuki 2016 [157]	972	Patients with acute HF	78.0	5.6	3.4–10.5	yes
Suzuki 2017 [158]	1079	Acute MI patients	67	3.7	4.6–6.4	yes
Schugar 2017 [120]	102	Patients with elective cardiac risk factor evaluation with T2D	55.9	4.8	3.3–7.7	yes
	333	Patients with elective cardiac risk factor evaluation without T2D	49.7	3.2	2.2–5.1	NA
Tang 2017 [159]	1216	Patients with T2D who underwent elective diagnostic coronary angiography	64.4	4.4	2.8–7.7	yes
	300	Apparently healthy controls	53.6	3.6	2.3–5.7	yes
Nie 2018 [160]	622	Hypertensive stroke patients	62.2	2.5	1.6–4.0	yes
	622	Matched controls	62.2	2.3	1.4–3.7	NA
Jaworska 2019 [161]	19	Cardiovascular patients qualified for aortic valve replacement	74.5	5.5 ± 0.6	--	No
	9	Healthy control	38.9	3.6 ± 0.4	--	NA
Reiner 2019 [162]	859	Venous thromboembolism patients	75.0	--	2.28–6.57	U-shaped optimum level at 4 µM
Hai 2015 [163]	7	Hemodialysis patients	Not indicated	(mean 77 ± 26)	Not indicated	Not indicated
	6	Control subjects	Not indicated	(mean 2 ± 1)	Not indicated	NA
Kaysen 2015 [164]	235	Patients new to hemodialysis	61.8	43	27.5–66.6	no
	NA	Two commercially available pooled control samples	Not indicated	(mean 1.41 ± 0.49)	NA	NA
Tang 2015 [165]	521	CKD patients	70	7.9	5.2–12.4	yes
	3166	Control subjects	62	3.4	2.3–5.3	NA
Missailidis 2016 [166]	56	CDK 3‒4 patients	42	14.6 *	5.6–71.2	yes
	55	CDK 5 patients on hemodialysis starting renal replacement	74	73.5 *	26.4–191.0	
	80	Controls	62	5.8 *	3.1–13.3	NA
Al-Obaide 2017 [167]	20	Diabetic CKD patients	64.4	12.5 (mean 20.2)	9.9–22.9	yes
	20	Healthy controls	54.3	1.2 (mean 2.4)	0.68–4.5	NA
Shafi 2017 [168]	1232	Hemodialysis patients, 35% Caucasians and 65% African American	58	White 87 Black 88 (total mean 101.9 ± 63.9)	63–120 62–125	Yes in Caucasians No in African Americans

IQR, interquartile range; * high level in healthy control group; ** 98 T2D patients with a mean TMAO plasma level of 8.6 ± 12.2 µM, compared to 5.4 ± 5.2 μM among 185 T2D free subjects.

**Table 2 nutrients-12-01330-t002:** Examples of preclinical evidence for TMAO as a cardiometabolic risk factor ***.

Literature Source	Test System	Treatment	Outcome	Model and Supplementation	Daily Dosage of TMAO or Precursors *	Plasma TMAO Level * (or Control Level)
Wang 2011 [115]	Atherosclerosis-prone male and female mice (apoE−/−)	Control or diet (around 0.08% choline) or diet with 0.5% or 1.0% additional choline or 0.12% TMAO; from weaning–20 weeks of age (16 weeks treatment)	Choline supplementation groups resulted in increased TMAO levels, with correlation between plasma TMAO and atherosclerotic plaque size. Female mice had higher FMO activity and higher TMAO levels upon supplementation, however plaques were observed similarly in both genders, introducing a confounding factor. Parallel examination of plasma cholesterol, triglycerides, lipoproteins, glucose levels, and hepatic triglyceride content in the mice failed to show significant increases that could account for the enhanced atherosclerosis.	apoE−/− mice; high choline/TMAO supplementation	800 or 1600 mg choline/kg ** or 192 mg TMAO/kg **	Males/females *w/0.5% choline:* 13 µM/140 µM; *w/1% choline*: 24 µM/215 µM *w/0.12% TMAO*: 23 µM/70 µM (5 µM/10 µM controls)
	Atherosclerosis-prone mice (apoE−/−) Gender not clearly indicated (Figure S7b)	Control or diet or diet with 1.0% additional choline or 0.12% or TMAO or 1% betaine; for at least 3 weeks half of them +antibiotics, for 3 weeks	Mice supplemented with either choline, TMAO, or betaine showed enhanced levels of scavenger receptors CD36 and SR-A1 (which bind to lipoproteins), markers for activated macrophages. Increase in scavenger receptors was inhibited by antibiotics. Mice on the control diet showed modest foam cell formation; 1% choline supplemented diet resulted in enhanced lipid-laden macrophages (foam cells). Foam cell formation was inhibited by antibiotics.	High choline or betaine supplementation	1600 mg choline/kg ** or 192 mg TMAO/kg ** or 1600 mg betaine/kg **	50 µM (4 h after challenge w/d9(trimethyl)-choline (baseline 0 µM) TMAO conc. after other supplementation not indicated
Koeth 2013 [94]	Atherosclerosis-prone female mice (apoE−/−)	1.3% L-carnitine in drinking water (or control water) +/− antibiotics; standard chow, for at least 4 weeks	Significant increase in plasma TMA and TMAO in the L-carnitine group without antibiotics; negligible concentrations in the other groups. L-carnitine did not increase other pro-atherogenic biomarkers (plasma lipids, lipoproteins, glucose, or insulin levels) in the plasma. 1.8-fold increase of aortic root lesion size in the L-carnitine group without antibiotics compared to the control, other groups similar to control. The difference between supplemented group and control was only 3 mice.	apoE−/− mice; high L-carnitine supplementation	2080 mg L-carnitine/kg **	130 µM (10 µM)
	Atherosclerosis-prone female mice (apoE−/−)	1.3% L-carnitine diet or 1.3% choline in diet; +/− antibiotics; standard chow, for 8 weeks	TMA/TMAO production abolished with antibiotics. TMAO was shown to inhibit reverse cholesterol transport in aortic cells.	apoE−/− mice; high L-carnitine or choline supplementation	2080 mg L-carnitine/kg ** 2080 mg choline/kg **	190 µM-w/carnitine 110 µM-w/choline (10 µM)
	Atherosclerosis-prone female mice (apoE−/−)	Supplementation of diet with 0.12% TMAO for 4 weeks	TMAO was shown to inhibit reverse cholesterol transport and reduce the expression of Cyp7a1 (enzyme involved in bile acid synthesis and cholesterol metabolism).	apoE−/−; TMAO supplementation	192 mg TMAO/kg **	35 µM (9 µM)
Ufnal 2014 [188]	Rats gender not relevant	Osmotic pump infusion with saline, TMAO, low-dose Angiotensin II, or both	TMAO did not affect blood pressure in normotensive animals. However, it prolonged the hypertensive effect of Angiotensin II.	High TMAO/angiotensin	Osmotic pump infusion with TMAO	TMAO: 58 µM (0.57 µM)
Seldin 2016 [182]	Female LDLR(-/-) mice	“Chow with 1.3% choline provided ad libitum in drinking water” or control	Aortas of LDLR(-/-) mice fed a choline diet showed elevated inflammatory gene expression compared with controls.	LDLr(-/-); high choline supplementation	2080 mg choline/kg **	55 µM (9 µM)
Tang 2015 [165]	Male mice	Diet with 1.0% choline or 0.12% TMAO, or control for 6 weeks or 16 weeks	Elevated TMAO levels were associated with increases in tubulointerstitial fibrosis, collagen deposition and phosphorylation of Smad3 (regulator of the pro-fibrotic TGF-β/Smad3 signaling pathway). TMAO-fed and choline-fed mice experienced increased kidney injury marker-1. After 16 weeks increased serum cystatin C levels compared to chow-fed mice were observed. Altogether, dose-dependent relationships were noted between plasma TMAO levels and monitored indices of renal histopathological and functional impairment.	High choline or TMAO diet	1600 mg choline/kg ** or 192 mg TMAO/kg **	100 µM-w/choline 40 µM-w/TMAO (5 µM control)
Zhu 2016 [183]	Female mice	Diet supplemented with either 0.12% TMAO or 1% choline or control; antibiotic control for 6 weeks	Choline or TMAO supplementation led to increased TMAO levels. Ex vivo platelet aggregation (induced by ADP stimulation) was significantly increased in these groups; antibiotics suppressed this effect for the choline supplemented mice (not the TMAO supplemented mice).	High choline supplementation	192 mg TMAO/kg ** 1600 mg choline/kg	Not indicated
Boini 2017 [184]	Male mice with partially ligated carotid artery	Osmotic pump infusion with TMAO (dosage not apparent) or control for 2 weeks post ligation	Mice with partially ligated carotid artery and infused with TMAO for 2 weeks had increased inflammasome formation.	Partially ligated carotid artery; TMAO infusion	Osmotic pump infusion with TMAO (dosage not apparent)	Not indicated
Li 2017 [189]	Male Fischer 344 rats (Most Fischer 344 rats older than 2 years exhibit small local areas of nephritis; less than 25% show severe nephritis [190])	Young or old (22 months) rats treated with 1% DMB in the drinking water for 8 weeks; controls	Compared with the young control group, the old control group had higher plasma TMAO levels. In both age groups DMB reduced TMAO levels. Old aortae exhibited increased expression of proinflammatory cytokines and superoxide production and decreased expression of endothelial nitric-oxide synthase (eNOS), all of which were restored by DMB treatment.	Old age: causes renal problems with these type of rats [190]	DMB treatment to inhibit TMAO formation	Old rats: 14.3 µM (reduced by DMB to 6.0 µM) (young rats 6.4 µM; reduced by DMB to 3.9 µM)
Yang 2019 [191]	Coronary ligation to induce myocardial infarction (MI), or sham operation in male mice	Mice were fed a control diet, high choline diet (1.2%) or/and DMB (choline analogue and inhibitor) diet or a TMAO diet (0.12%) starting 3 weeks before MI; treatment for one more week after MI	Cardiac fibrosis increased with 0.12% TMAO or 0.24% TMAO, but not with 0.06% TMAO. In the MI model, TMAO or choline supplementation led to decreased cardiac function and cardiac fibrosis. DMB inhibited the TMAO-effect. Choline or TAMO treated MI animals transformed fibroblasts into myofibroblasts and activated the TGF-βRI/Smad2 pathway, indicating cardiac fibrosis.	MI, choline or high TMAO or choline supplementation	1920 mg choline/kg or 192 mg TMAO/kg **	Before MI: 42.4 µM w/TMAO (8.8 µM control) before MI: 91.6 µM w/choline (6.1 µM control; 27.8 µM w/choline + DMB)
Chen 2019 [192]	Female mice	Mice treated with 1.3% L-carnitine in drinking water (drinking volume and body mass provided), flavonoids (or oolong tea)	Mice treated with L-carnitine significantly increased plasma TMAO levels compared to control. L-carnitine also increased inflammation markers. TMAO was remarkedly reduced by flavonoids. Antibiotics strongly reduced TMAO production. Inflammation markers (TNF-α, E-selectin, and VCAM-1) were reduced by flavonoids or oolong tea or antibiotics.	High L-carnitine → high TMAO	2941 mg L-carnitine in L-carnitine group; 2657 mg L-carnitine in flavonoid group; 2424 mg L-carnitine in antibiotics group	400 µM w/L-carnitine (26 µM with no diet; 322 µM with flavonoids; 13 µM with antibiotics)
Li 2019 [193]	Male rats with coronary ligation to induce MI or sham operation	MI and sham rats treated with either vehicle (tap water) or 1.0% DMB in tap water, for 8 weeks	Plasma TMAO levels were elevated in vehicle-treated MI rats compared with vehicle-treated sham rats; plasma TMAO levels were reduced in DMB-treated MI rats. Manifestations of MI-induced heart failure were improved in DMB-treated MI rats. Elevated plasma TMAO levels went along with increased proinflammatory IL-8 plasma level in MI groups. In sham rats, DMB treatment reduced plasma TMAO but did not alter other parameters.	Coronary ligation → myocardial infarction	DMB treatment to inhibit TMAO formation	30 µM w/o DMB (Sham: 12 µM w/o DMB) (MI & DMB: 10 µM) (Sham & DMB: 3 µM)
Chen 2019 [194]	Male mice	1% cholesterol diet with/without 1% choline	Plasma TMAO levels increased about 4-fold with diet + choline; increased mRNA of cholesterol uptake and secretion genes (Abcg5 and g8, Ldlr); no difference in bile acid composition.	High cholesterol diet, high choline	1600 mg choline/kg **	7.7 µM (1 µM)
	Male mice	1% cholesterol diet and supplemented with low dose (0.12%) or high dose (0.3%) TMAO	No pathological differences in liver tissue; cholesterol concentration in gallbladder bile increased with TMAO, more apparent at high dose; increased mRNA of cholesterol uptake and secretion genes (Abcg5 and g8, Ldlr, Srb1).	High cholesterol diet, TMAO supplementation	192 mg TMAO/kg or 480 mg TMAO/kg	Not indicated
	Gallstone-susceptible AKR/J male mice (biliary cholesterol hypersecretion)	Lithogenic diet supplemented with 0.3% TMAO or not supplemented (control).	With TMAO, the incidence of gallstones rose to 70%, compared no gallstones in the control mice. TMAO also induced increased hepatic Abcg5 and g8 expression.	Lithogenic diet; gallstone-susceptible AKR/J mice (cholesterol hypersecretion); high TMAO supplementation	480 mg TMAO/kg	23,3 µM (1 µM lithogenic diet)

* Plasma TMAO levels consecutive to the indicated intervention. In the case of multiple dosing levels, it corresponds to the minimum dosage for unfavourable effects and means that indicated dosages resulted in significant, relevant effects. Several concentrations are approximations derived from graphical illustrations, e.g., bar graphs (numerical values were not always published). Some baseline values are rather high, possibly indicating alternate methodology (see also section on methodology). ** Calculated from dietary intake as follows: The average daily consumption of feed and water for an adult 25 g mouse is 3–5 g (averaged to 4 g) and 4 mL (= 4 g) respectively [195]; 0.12% TMAO = 1,2 mg/g; → average daily dosage with 4 g intake: 4.8 mg. Male adult apoE−/− mice [196] weight around 30 g. Female adult apoE−/− mice weight around 20 g. It needs to be assumed that males eat and drink more than females. Thus, taking an average mouse weight of 25 g will be a good approximation for both genders. → Daily dosing would be approximately 4.8 mg/25 g body mass (bm) = 4.8 mg/0.025 kg bm → 192 mg TMAO/kg bm. 1% choline = 10 mg/g; → average daily dosage with 4 g intake: 40 mg → daily dosing 40 mg/25 g bm= 40 mg/0.025 kg bm → 1600 mg choline/kg bm. 1.3% L-carnitine = 13 mg/g; → average daily dosage with 4 g intake: 52 mg → daily dosing 52 mg/25 g bm = 52 mg/0.025 kg bm → 2080 mg L-carnitine/kg bm. Other calculations: if 352 mg/kg L-carnitine is equivalent to 2000 mg/day in humans [185], 2080 mg/kg L-carnitine would be equivalent to 11818 mg/day in humans (factor 5.9). *** Literature examples were randomly selected in the sequence as discovered during the search. Further literature appears to be similar in presenting adverse effects coupled to distress of experimental animals and rather high TMAO concentrations, e.g., [197,198,199,200,201].

**Table 3 nutrients-12-01330-t003:** Preclinical evidence for cardio-protective or neutral effects of TMAO.

Literature Source	Test System	Treatment	Outcome	Distress Factors	Daily Dosage of TMAO or Precursors *	Plasma TMAO Level * (or Control Level)
Mayr 2005 [114]	Male and female apoE−/− and apoE+/+mice on normal chow diet	Proteomics and metabolomics to identify protein and metabolite changes in vessels	No significant difference in TMAO concentration in the aortas of 18-month-old apoE−/− and ApoE+/+mice. Lesion formation in apoE−/− mice due to increase in oxidative stress, not to increased TMAO.	apoE−/− mice	none	Females 0.06 Males 0.25 (units not stated, controls not stated)
Martin 2009 [202]	Male hamsters	Hyperlipidemic diet (normal diet plus 100 g/kg fat for 5 weeks + 200 g/kg for 12 weeks; fat as anhydrous butter or cheese) or controls; (1)H NMR-based metabonomics	VLDL lipids, cholesterol, and N-acetylglycoproteins had the best correlation to onset of atherosclerosis. TMAO was found to be negatively associated with atherogenesis.	High-fat diet	none	Absolute concentrations not determined (only relative ones; personal communication)
Gao 2014 [203]	Male mice	Control, diet with 25% fat +/− 0.2% TMAO for 4 weeks	Dietary TMAO increased fasting insulin levels and insulin resistance and exacerbated impaired glucose tolerance and MCP-1 mRNA (pro-inflammatory cytokine) in HFD-fed mice. IL-10 mRNA (anti-inflammatory cytokine) in adipose tissue was (more than already by the high-fat diet) decreased by TMAO. However, the increase in atherosclerosis associated with a high-fat diet was prevented by TMAO, suggesting a protective effect of TMAO with regard to atherosclerosis.	High-fat diet; TMAO supplementation	320 mg TMAO/kg **	17.5 µM (normal chow 11.9 µM; high-fat chow 12 µM)
Shih 2015 [118]	Male mice, transgenic FMO_3_ overexpression	Transgenic compared to control mice, supplemented with water containing 1.3% choline chloride for 6 weeks	FMO_3_ overexpression caused a 75% increase in plasma TMAO levels and increased hepatic and plasma lipids.	FMO_3_ overexpression; high choline supplementation	FMO3 overexpression; 2080 mg choline/kg **	16 µM (9 µM for control transgene)
Shih 2015 [29,118]	Male hyperlipidemic mouse “E3L Tg” with transgenic FMO_3_ overexpression	Transgenic compared to control mice; low-fat or high-fat/1% cholesterol chow for 16 weeks	Increased plasma TG, VLDL/IDL/LDL, and unesterified cholesterol with both diets, increased glucose and insulin levels, increased levels of TG, TC, and phosphatidylcholine in the VLDL plasma fractions with high-fat diet. Small increase (20%) in atherosclerotic lesion size compared to knockdown mouse (see above); it is implausible to explain the effect with the slight change in TMAO levels.	Hyperlipidemic mouse with transgenic FMO_3_ overexpression; high-fat/high-cholesterol chow	High-fat diet	w/high-fat/cholesterol: 2.6 µM (2.2 µM for control transgene) (difference = trend)
Collins 2016 [185]	Male apoE−/− mice expressing human cholesteryl ester transfer protein (hCETEP)	12 week treatment with L-carnitine (87 mg/kg and 352 mg/kg; equivalent to 500 and 2000 mg/day in humans); (and/or methimazole =FMO inhibitor)	High doses of L-carnitine resulted in a significant increase in plasma L-carnitine and TMAO levels. Plasma lipid and lipoprotein levels did not change. Aortic root analysis showed significant decrease in lesion size with L-carnitine treatment and high TMAO compared to the control. Significantly lower levels of lesion were found with elevated plasma TMAO (>0.05 ppm) compared to the low plasma TMAO (<0.05 ppm). TMAO may be protective against atherosclerosis development.	apoE−/− mice (plus hCETEP)	352 mg L-carnitine/kg	0.2 ppm = 2.7 µM (0.08 ppm = 1.07 µM)
Empl 2015 [204]	Male Fischer 344 rats	Daily 0, 0.1, 0.2 or 0.5 g/L L-carnitine (0; 70; 141; 352 mg/kg) via drinking water for one year	L-carnitine did not cause any preneoplastic, atherosclerotic, or other lesions.	None	352 mg L-carnitine/kg (highest dosage)	See below
Weinert 2017 [205]	Male Fischer 344 rats from study [204]	Daily 0, 0.1, 0.2 or 0.5 g/L L-carnitine (0; 70; 141; 352 mg/kg) via drinking water for one year	High dose L-carnitine resulted in tenfold higher plasma TMAO concentration compared to the control (25.0 μM). Supplementation did not cause changes in the plasma metabolome.	None	352 mg L-carnitine/kg (highest dosage)	25.0 μM (2.5 µM for control)
Huc 2018 [206]	Male spontaneously hypertensive rats (SHR) with pressure-overloaded hearts, 2 age groups	TMAO 333 mg/L with drinking water, water controls and normotensive rat controls for 9 weeks; metabolic cage for 2 days at end of study	Chronic, low-dose trimethylamine oxide (TMAO) treatment increased plasma TMAO by 4 to 5-fold and reduced plasma NH2-terminal pro-B-type natriuretic peptide and vasopressin, left ventricular end-diastolic pressure, and cardiac fibrosis. TMAO may be beneficial for reduction of hypertension.	Hypertension	37 (16 weeks)/32 mg (56 weeks) TMAO/kg (personal communication)	16 weeks: 37.3 µM (SHR control: 8.8 µM) (normal control: 6.3 µM) 56 weeks: 40.9 µM (SHR control: 8.1 µM) (normal control: 5.2 µM)
Lindskog Jonsson 2018 [207]	Male germ-free or conventionally raised apoE−/− mice	Western diet alone or supplemented with 1.2% choline for 12 weeks	Conventionally raised mice had smaller aortic lesions and lower plasma cholesterol levels on a chow diet compared to a Western diet; choline supplementation increased plasma TMAO levels in conventionally raised mice but not in germ-free mice. However, choline supplementation did not affect the size of aortic lesions or plasma cholesterol levels; increased plasma TMAO levels observed in conventionally raised apoE−/− mice fed a choline-supplemented diet did not correlate with aortic lesion size. The microbiota was required for TMAO production from dietary choline, but this process could not be linked to increased atherosclerosis. Choline supplementation reduced body weight and epididymal fat weight in conventionally but not germ-free mice when fed Western diet.	Western diet/germ-free existence; apoE−/− mice	1920 mg choline/kg **	Conventionally raised: Western diet +choline: 8 µM Chow +choline: 21 µM (Western diet: 0.5 µM Chow: 1 µM) No TMAO in germ-free mice
Zhao 2019 [208]	Male rats with steatohepatitis induced by high-fat high-cholesterol diet	16-wk high-fat high-cholesterol (HFHC) diet feeding; daily TMAO (120 mg/kg/day) by oral gavage for 8 weeks	Hepatic and serum levels of cholesterol were both decreased by TMAO treatment in rats on HFHC diet. TMAO treatment also downregulated cholesterol influx-related Niemann-Pick C1-like 1 (intestinal cholesterol absorption transporter) and upregulated cholesterol efflux-related ABCG5/8 in the small intestine; thus, TMAO inhibits intestinal cholesterol absorption. Gut microbiota analysis showed that TMAO could alter the gut microbial profile and restore the diversity of gut flora. TMAO also ameliorated hepatic ER stress and cell death under cholesterol overload, thereby attenuating HFHC diet-induced steatohepatitis in rats.	Steatohepatitis induced by high-fat high-cholesterol diet	120 mg TMAO/kg	Not determined (personal communication)
Aldana-Hernandez 2019 [209]	Male Ldlr-/- mice and male apoE−/− mice	40% high-fat diet for atherosclerosis induction; control (0.1% choline) or supplemented with 1% choline or 0.9% betaine or 0.2% TMAO; for 8 or 16 weeks control diet (0.1% choline) or diet for ≤28 weeks supplemented with 1% choline or 0.9% betaine or 0.12% TMAO for 12 or 28 weeks	In LDLr-/- mice, dietary supplementation for 8 wk with choline or TMAO increased plasma TMAO concentrations by 1.6-and 4-fold, respectively. After 16 wk, there was a 2-fold increase in plasma TMAO after dietary TMAO supplementation. In apoe-/- mice, dietary supplementation with choline, betaine, or TMAO for 12 wk did not increase plasma TMAO concentrations. However, choline and TMAO supplementation for 28 wk significantly increased plasma TMAO concentrations by 1.8-and 1.5-fold, respectively. Atherosclerotic lesion size was not altered by any of the dietary interventions, irrespective of mouse model.	Ldlr-/- mice (40% high-fat diet) apoE−/− mice	1600 mg choline/kg (1%) ** 1440 mg betaine/kg (0.9%) ** 320 mg TMAO/kg (high TMAO dosage of 0.2%) ** 192 mg TMAO/kg (low TMAO dosage of 0.12%) **	LDLr-/- mice (8 weeks/16 weeks): w/choline: 1.3 µM/1.8 µM w/betaine: 0.9 µM/1.1 µM w/TMAO: 2.5 µM/1.1 µM (0.5-1.1 µM for controls) apoE−/− mice (12 weeks/28 weeks): w/choline: 1.3µM/1.6 µM w/betaine: 0.8 µM/0.8 µM w/TMAO: 0.9 µM/1.3 µM (1.0 µM/0.8 µM for controls)

* Plasma TMAO level consecutive to the indicated intervention. In the case of multiple dosing levels, it corresponds to the maximum dosage for beneficial/neutral effects and means that they are dosages that result in significant, relevant effects. Several concentrations are approximations derived from graphical illustrations, e.g., bar graphs (numerical values were not always published). Some baseline values are rather high, possibly indicating alternate methodology (see also section on methodology). ** See footnote ** of Table 2 for calculation.

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
