# Peer review of "Trimethylamine N-Oxide in Relation to Cardiometabolic Health—Cause or Effect?"

_nutrients, 2020, doi:10.3390/nu12051330_

Round 1

Reviewer 1 Report

In my opinion the paper entitled ‘Trimethylamine N-oxide in Relation to Cardiometabolic Health—Cause or Effect?’ fits well in the scope of the Journal and could be interesting for the readers of Nutrients. The structure of the manuscript is well organize. The authors paid special attention on epidemiological and preclinical evidence on the impacts of TMAO in cardiometabolic diseases, particularly as related to gut microbiota. After reading the manuscript, however, I have several issues I feel need to be addressed and would welcome the author’s comments on these.

Major comments:

  1. P3: The authors should mention in a few sentences that the relationship between TMA, TMAO, and DMA. TMAO can be converted to TMA. TMAO can also be degraded to DMA. Thus, levels of TMAO and TMAO can be dynamically altered. Please also consider and discuss the impact of urinary TMAO and the TMAO-to-TMA ratio as a marker in clinical study.
  2. Line 226: TMAO variability. Authors discussed TMAO variability mainly in the blood. How about urinary TMAO variability? Another possibility of TMAO variability come from methodology. Please discuss. Will age and sex affect TMAO levels?
  3. Line 239: 3.1.1. TMAO-variability through FMO. The authors should describe in a more comprehensive way about different FMO isoforms in different organs or possible organ-specific effects of TMAO. For example, it is known that serum TMAO concentrations are increased in chronic kidney disease patients, which may be due in part to increased hepatic FMO3-mediated TMAO formation. What is the role of other FMOs in the kidney and vessels? And their role in CVD in patients with CKD?
  4. Line 289: 3.1.3. TMAO-variability by microbiome variability. There are many microbes involved in the synthesis of TMA and TMAO, as well as in conversion of TMAO to TMA. I doubt about the amount of information that demonstrated in the current review is too little. (References: Rath et al., Uncovering the trimethylamine-producing bacteria of the human gut microbiota. Microbiome 2017;5:54; Hoyles et al., Metabolic retroconversion of trimethylamine N-oxide and the gut microbiota. Microbiome 2018;6:73.)
  5. Line 435: 5.1. TMAO in cell culture. Any new evidence about TMAO receptor?

Minor comments:

  1. Line 13: “TMAO plasma concentrations” should be “plasma TMAO concentrations”.
  2. Line 54: Again, please replace “TMAO plasma levels” to “plasma TMAO levels”.
  3. Line 329: TMAO and cardiometabolic disease. Almost all reports come from adults. Adding data from pediatric population will be very helpful.
  4. Line 329: Typo: “But what determines TMAO levels?”
  5. Line 393: “may be just a feedback mechanism to counter disease progression in light of its role as a molecular chaperone”. Please cite the reference.
  6. Line 496: In addition to atherosclerosis, DMB has been shown to prevent hypertension (Hsu et al., Mol Nutr Food Res 2019).
  7. Line 547: section on methodology? There is no methodology section in this review.
  8. Line 621: “On the other hand, several animal experiments using lower doses of TMAO and its precursors did show protective effects.” Please cite the references.
  9. Line 634: “Such activitiy may include chaperone like functions of TMAO” seems overstated. Also, “activitiy” is a typo.
  10. Line 638: The literature search up to November 11, 2019. The new update will help readers find the most up-to-date results.

I hope that the enclosed comments will be of help to the authors. 

Author Response

In my opinion the paper entitled ‘Trimethylamine N-oxide in Relation to Cardiometabolic Health—Cause or Effect?’ fits well in the scope of the Journal and could be interesting for the readers of Nutrients. The structure of the manuscript is well organize. The authors paid special attention on epidemiological and preclinical evidence on the impacts of TMAO in cardiometabolic diseases, particularly as related to gut microbiota. After reading the manuscript, however, I have several issues I feel need to be addressed and would welcome the author’s comments on these.

Reply: We sincerely thank Reviewer 1 for the global appreciation of our investigation, as well as for all the valuable comments and suggestions provided in the following lines, which have greatly improved the first version of the manuscript. We have addressed all them in each of the following points, as well as in the manuscript, when required. Please find below the itemized responses to all Reviewer’s 1 comments.

Major comments:

  1. P3: The authors should mention in a few sentences that the relationship between TMA, TMAO, and DMA. TMAO can be converted to TMA. TMAO can also be degraded to DMA. Thus, levels of TMAO and TMAO can be dynamically altered. Please also consider and discuss the impact of urinary TMAO and the TMAO-to-TMA ratio as a marker in clinical study.

Reply: This information is now described in the beginning of the paragraph 2 “Where does the TMAO come from?” Additional we concluded this paragraph by the following statement: Therefore, the plasma TMAO levels are influenced by the TMA formation and its degradation as well as secretion rate of TMA and TMAO. Perhaps it would be more relevant to consider plasma TMA/TMAO ratio as a marker rather than just TMAO levels.

  1. Line 226: TMAO variability. Authors discussed TMAO variability mainly in the blood. How about urinary TMAO variability? Another possibility of TMAO variability come from methodology. Please discuss. Will age and sex affect TMAO levels?

Reply to TMAO variability based on urinary levels: We recognized the importance of assessing urine TMAO levels per the previous comment. However, most of the studies reported only plasma levels. we tried to clarify through the text whether the TMAO is measured in the plasma or in the urine. Further, we added the following comment under “fish intake”:

As it is the case for plasma TMAO, urinary TMAO levels are also variable and observed in particular with high fish intake in humans (Yin, 2020). However, in most publications, only serum TMAO levels are considered as a marker. In addition, we added the following statement to the conclusion: Another factor in such variability is related to the methodology itself. Indeed, most studies reported plasma TMAO only without considering TMA or urinary secretion. Perhaps it would be more relevant to consider plasma TMA/TMAO ratio as a marker.

Regarding sex and age, an additional paragraph has been added:

TMAO variability with age and sex

Older subjects were found to have high levels of TMAO. In a study comparing 168 young, 118 middle aged and 141 elderly adults, mean TMAO levels were around 2.5 µM, 4.8 µM and 10 µM, respectively. However, inter-individual variability was also observed within the same demographic. Other reported higher TMAO levels in older people (Li, 2018). Animal testing reported similar observations in old mice (Ke 2018)

Gender differences may play a role in the observed TMAO levels and has been reported between men and women. Indeed, differences in TMAO levels are possibly due to differences in food consumption patterns levels (Razavi, 2019). However, the gender effect was not reported by others (Bennett, 2013). Animal models reported effects of gender on TMAO levels related to differences in the FMO expression. This is discussed in details in the following paragraph

  1. Line 239: 3.1.1. TMAO-variability through FMO. The authors should describe in a more comprehensive way about different FMO isoforms in different organs or possible organ-specific effects of TMAO. For example, it is known that serum TMAO concentrations are increased in chronic kidney disease patients, which may be due in part to increased hepatic FMO3-mediated TMAO formation. What is the role of other FMOs in the kidney and vessels? And their role in CVD in patients with CKD?

Reply: The following paragraph has been added to the FMO variability:

Variability in FMO3 expression was reported in patients with CKD, leading possibly to a variability in renal clearance of TMAO159. These effects could be the results of activation in FMOs elicited by octylamine and uremic serum161. Lower clearance in TMAO coincided with increased FMO3 expression in a CKD mice model160. High plasma TMAO levels in CKD patients may contribute to the observed damage in blood vessels161. In pediatric CKD patients, plasma TMAO, DMA and TMA levels are also increased, while the level of these compounds was decreased in the urine162. Microbiome was found to be altered in children with CKD where Cyanobacteria, Subdoligranulum, Faecalibacterium, Ruminococcus, and Akkermansia abundance are affected162. In a different study with CKD children, the Prevotella genus was reduced, while Lactobacillus and Bifidobacterium were increased 136,127. More recently, Rath and colleagues reported the role of diet and age in the abundance of the TMA-producing bacteria and how the diet and supplementation can modulate the abundance of the microbiota leading to the formation of TMA (Rath, 2020). In particular, bacteria from Clostridium XIVa and Eubacterium sp., containing choline TMA-lyase genes, and Gammaproteobacteria, containing L-carnitine oxygenase genes have been identified in in the gut microbiota (Rath, 2017)

  1. Line 289: 3.1.3. TMAO-variability by microbiome variability. There are many microbes involved in the synthesis of TMA and TMAO, as well as in conversion of TMAO to TMA. I doubt about the amount of information that demonstrated in the current review is too little. (References: Rath et al., Uncovering the trimethylamine-producing bacteria of the human gut microbiota. Microbiome 2017;5:54; Hoyles et al., Metabolic retroconversion of trimethylamine N-oxide and the gut microbiota. Microbiome 2018;6:73.)

Reply: the microbiome is now discussed and added to the section on the FMO variability (please above). The effect of reconversion was added to the paragraph discussing the TMAO formation (Hoyles, 2018)

  1. Line 435: 5.1. TMAO in cell culture. Any new evidence about TMAO receptor?

Reply: This paragraph has been added to the cell culture section: Recently, TMAO was found to bind to endoplasmic reticulum stress kinase PERK (EIF2AK3), selectively activating the PERK branch of the unfolded protein response. This induced the activation of the transcription factor FoxO1, which has been described to suppress lipogenesis (Chen, 2019).

Minor comments:

  1. Line 13: “TMAO plasma concentrations” should be “plasma TMAO concentrations”.

We corrected it.

  1. Line 54: Again, please replace “TMAO plasma levels” to “plasma TMAO levels”.

We corrected it.

  1. Line 329: TMAO and cardiometabolic disease. Almost all reports come from adults. Adding data from pediatric population will be very helpful.

We added other demographics through the text (pediatrics and older subjects)

  1. Line 329: Typo: “But what determines TMAO levels?”

We deleted it.

  1. Line 393: “may be just a feedback mechanism to counter disease progression in light of its role as a molecular chaperone”. Please cite the reference.

Following this comment, we cited the following reference: Shepshelovich J, Goldstein-Magal L, Globerson A, Yen PM, Rotman-Pikielny P, Hirschberg K. Protein synthesis inhibitors and the chemical chaperone TMAO reverse endoplasmic reticulum perturbation induced by overexpression of the iodide transporter pendrin. J Cell Sci. 2005 Apr 15;118(Pt 8):1577-86.

  1. Line 496: In addition to atherosclerosis, DMB has been shown to prevent hypertension (Hsu et al., MolNutr Food Res 2019).

We added a relevant sentence and cited the suggested study.

  1. Line 547: section on methodology? There is no methodology section in this review.

The methodology section is described at the end of the manuscript under “Literature search methods” and on the figure 3.

  1. Line 621: “On the other hand, several animal experiments using lower doses of TMAO and its precursors did show protective effects.” Please cite the references.

We cited the references: 154, 165, 166, 167

  1. Line 634: “Such activitiy may include chaperone like functions of TMAO” seems overstated. Also, “activitiy” is a typo.

We agree with the Reviewer 1 and we deleted this sentence.

  1. Line 638: The literature search up to November 11, 2019. The new update will help readers find the most up-to-date results.

Additional literature published after November 11 2019 have been added

I hope that the enclosed comments will be of help to the authors._

Reviewer 2 Report

The authors have written an extensive and detailed review on the cause/effect role of TMAO in cardiometabolic diseases, including a complete revision on the role of TMAO as a biomarker or as a protecting molecule against cardiometabolic risk factor. However, several points should be addressed:

Authors should specify where increased/decreased "TMAO levels" are found (serum, plasma, urine) throughout the manuscript in order to avoid confusion. 

Please revise the meaning of a sentence in page 3, line 113: "Therefore, dietary supplementation with L-carnitine can contribute to dietary intake, particularly among vegetarians, and people with chronic diseases associated with aging". 

A reference should be added to the sentence "In addition, L-carnitine supplementation reduces oxidative stress, inflammation and necrosis in many tissues including the cardiac tissue." in page 4, line 127.

Also, it is necessary to indicate in which organism or type of study some of the mentioned findings took place, there are several examples of this missing information in the review (for example page 4, line 130 "L-carnitine supplementation was shown to normalize plasma lipid levels" (in humans, adults, mice...??).

The choline paragraph in section 2 could benefit from some additional comments on the contradictory nature of choline being necessary and adequate for the maintenance of normal lipid metabolism, normal liver function, and normal homocysteine metabolism, but also associated with CVD in some cross-sectional and prospective studies. If choline does not affect TMAO plasma concentrations, perhaps it is not necessary to include the rest of the information... Please consider.

Information in point 2 could be summarized in a table.

Revise numbering of sections (I believe TMAO variability should be section 3.1).

From point 4 onwards, many sections are sequenced summaries of the cited works. Perhaps a table could be used to clearly indicate plasma TMAO concentration ranges found in controls and patients with disease, as explained in section 4. However, this one and some of the following sections could benefit from rewriting in a more summarized and processed manner in order to help the reader find similarities and differences between the cited studies.

Finally, several recent works have not been included and could definitely be part of this review:

  • Fu et al. Am J Clin Nutr. 2020
  • Warmbrunn et al. Expert Rev Endocrinol Metab 2020; 15(1):13-27
  • Roy et al. PLoS One. 2020 Jan 15;15(1):e0227482
  • Genoni et al. Eur J Nutr. 2019
  • Park et al. Nutr Metab Cardiovasc Dis. 2019 May;29(5):513-517

Author Response

The authors have written an extensive and detailed review on the cause/effect role of TMAO in cardiometabolic diseases, including a complete revision on the role of TMAO as a biomarker or as a protecting molecule against cardiometabolic risk factor. However, several points should be addressed:

Reply: We sincerely thank Reviewer 2 for the global appreciation of our investigation, as well as for all the valuable comments and suggestions provided in the following lines, which have greatly improved the first version of the manuscript. We have addressed all them in each of the following points, as well as in the manuscript, when required. Please find below the itemized responses to all Reviewer’s 2 comments.

Authors should specify where increased/decreased "TMAO levels" are found (serum, plasma, urine) throughout the manuscript in order to avoid confusion. 

Reply: We specified the biofluids in which TMAO was assessed throughout the manuscript.

Please revise the meaning of a sentence in page 3, line 113: "Therefore, dietary supplementation with L-carnitine can contribute to dietary intake, particularly among vegetarians, and people with chronic diseases associated with aging". 

Reply: We rephrased this sentence as follows: “Therefore, dietary supplementation with L-carnitine can help vegetarians, and people with chronic diseases associated with aging meet their daily requirements in L-carnitine”.

A reference should be added to the sentence "In addition, L-carnitine supplementation reduces oxidative stress, inflammation and necrosis in many tissues including the cardiac tissue." in page 4, line 127.

Reply: We added the following reference: Wang ZY, Liu YY, Liu GH, Lu HB, Mao CY. l-Carnitine and heart disease. Life Sci 2018; 194: 88-97.

Also, it is necessary to indicate in which organism or type of study some of the mentioned findings took place, there are several examples of this missing information in the review (for example page 4, line 130 "L-carnitine supplementation was shown to normalize plasma lipid levels" (in humans, adults, mice...??).

Reply: Following this comment we indicated that L-carnitine supplementation was shown to normalize plasma lipid levels in patients with coronary artery disease. We also indicated in other parts of the manuscript the types of organism and study.

The choline paragraph in section 2 could benefit from some additional comments on the contradictory nature of choline being necessary and adequate for the maintenance of normal lipid metabolism, normal liver function, and normal homocysteine metabolism, but also associated with CVD in some cross-sectional and prospective studies. If choline does not affect TMAO plasma concentrations, perhaps it is not necessary to include the rest of the information... Please consider.

Reply: We added the following sentences in order to make clearer that circulating choline may reflect what occurs it the mitochondria: “A disruption of mitochondrial choline oxidation to betaine as part of mitochondrial dysfunction may precede the development of cardiovascular diseases. Oxidative stress and inflammation may impair betaine-homocysteinemethyltransferase activity, resulting in accumulation of circulating choline (FASEB J 2015;29:418-32).” Choline may affect TMAO production via its conversion to TMA through the action of choline TMAO lyase.

Information in point 2 could be summarized in a table.

A summary table might be included. However, given the length of the manuscript, we suggest to keep the information as a text only

Revise numbering of sections (I believe TMAO variability should be section 3.1).

TMAO variability is section 3.1

From point 4 onwards, many sections are sequenced summaries of the cited works. Perhaps a table could be used to clearly indicate plasma TMAO concentration ranges found in controls and patients with disease, as explained in section 4. However, this one and some of the following sections could benefit from rewriting in a more summarized and processed manner in order to help the reader find similarities and differences between the cited studies.

Table 1 has been added

Finally, several recent works have not been included and could definitely be part of this review:

  • Fu et al. Am J ClinNutr. 2020
  • Warmbrunn et al. Expert Rev EndocrinolMetab 2020; 15(1):13-27
  • Roy et al. PLoS One. 2020 Jan 15;15(1):e0227482
  • Genoni et al. Eur J Nutr. 2019
  • Park et al. NutrMetabCardiovasc Dis. 2019 May;29(5):513-517 Reply: We included these recent works as suggested by Reviewer 2.
  •  

Round 2

Reviewer 1 Report

Authors answered questions and comments that I have addressed to them by revision of the previous version of manuscript and included corresponding changes to the revised manuscript. The manuscript has been significantly improved with the re-write. Thank you for your attention to these matters.
